# Learning Unsigned Distance Fields from Local Shape Functions for 3D Surface Reconstruction

## Abstract

Unsigned distance fields (UDFs) provide a versatile framework for representing a diverse array of 3D shapes, encompassing both watertight and non-watertight geometries. Traditional UDF learning methods typically require extensive training on large datasets of 3D shapes, which is costly and often necessitates hyperparameter adjustments for new datasets. This paper presents a novel neural framework, *LoSF-UDF*, for reconstructing surfaces from 3D point clouds by leveraging local shape functions to learn UDFs. We observe that 3D shapes manifest simple patterns within localized areas, prompting us to create a training dataset of point cloud patches characterized by mathematical functions that represent a continuum from smooth surfaces to sharp edges and corners. Our approach learns features within a specific radius around each query point and utilizes an attention mechanism to focus on the crucial features for UDF estimation. This method enables efficient and robust surface reconstruction from point clouds without the need for shape-specific training. Additionally, our method exhibits enhanced resilience to noise and outliers in point clouds compared to existing methods. We present comprehensive experiments and comparisons across various datasets, including synthetic and real-scanned point clouds, to validate our method's efficacy.

## 1 Introduction

3D surface reconstruction from raw point clouds is a significant and long-standing problem in computer graphics and machine vision. Traditional techniques like Poisson Surface Reconstruction[1] create an implicit indicator function from oriented points and reconstruct the surface by extracting an appropriate isosurface. The advancement of artificial intelligence has led to the emergence of numerous neural network-based methods for 3D reconstruction. Among these, neural implicit representations have gained significant influence, which utilize signed distance fields (SDFs) [2–8] and occupancy fields [9–12] to implicitly depict 3D geometries. SDFs and occupancy fields extract isosurfaces by solving regression and classification problems, respectively. However, both techniques require internal and external definitions of the surfaces, limiting their capability to reconstructing only watertight geometries. Therefore, unsigned distance fields [13–20] have recently gained increasing attention due to their ability to reconstruct non-watertight surfaces and complex geometries with arbitrary topologies.

Reconstructing 3D geometries from raw point clouds using UDFs presents significant challenges due to the non-differentiability near the surface. This characteristic complicates the development of loss functions and undermines the stability of neural network training. Various unsupervised approaches [17, 14, 19] have been developed to tailor loss functions that leverage the intrinsic characteristics of UDFs, ensuring that the reconstructed geometry aligns closely with the original point clouds. However, these methods suffer from slow convergence, necessitating an extensive network training time to reconstruct a single geometry. As a supervised method, GeoUDF [15] learns local geometric

priors through training on datasets such as ShapeNet [21], thus achieving efficient UDF estimation. Nonetheless, the generalizability of this approach is dependent on the training dataset, which also leads to relatively high computational costs.

In this paper, we propose a lightweight and effective supervised learning framework, *Losf-UDF*, to address these challenges. Since learning UDFs does not require determining whether a query point is inside or outside the geometry, it is a local quantity independent of the global context. Inspired by the observation that 3D shapes manifest simple patterns within localized areas, we synthesize a training dataset comprising a set of point cloud patches by utilizing local shape functions. Subsequently, we can estimate the unsigned distance values by learning local geometric features through an attention-based network. Our approach distinguishes itself from existing methods by its novel training strategy. Specifically, it is uniquely trained on synthetic surfaces, yet it demonstrates remarkable capability in predicting UDFs for a wide range of common surface types. For smooth surfaces, we generate training patches (quadratic surfaces) by analyzing principal curvatures, meanwhile, we design simple shape functions to simulate sharp features. This strategy has three unique advantages. First, it systematically captures the local geometries of most common surfaces encountered during testing, effectively mitigating the dataset dependence risk that plagues current UDF learning methods. Second, for each training patch, the ground-truth UDF is readily available, streamlining the training process. Third, this approach substantially reduces the costs associated with preparing the training datasets. We evaluate our framework on various datasets and demonstrates its ability to robustly reconstruct high-quality surfaces, even for point clouds with noise and outliers. Notably, our method can serve as a lightweight initialization that can be integrated with existing unsupervised methods to enhance their performance. We summarize our main contributions as follows.

- We present a simple yet effective data-driven approach that learns UDFs directly from a synthetic dataset consisting of point cloud patches, which is independent of the global shape.

- Our method is computationally efficient and requires training only once on our synthetic dataset. Then it can be applied to reconstruct a wide range of surface types.

- Our framework achieves superior performance in surface reconstruction from both synthetic point clouds and real scans, even in the presence of noise and outliers.

## 2 Related Work

**Surface reconstruction.** Reconstructing 3D surfaces from point clouds is a classic and important topic in computer graphics. The most widely used Poisson method [1, 22] fits surfaces by solving PDEs. These traditional methods involve adjusting the gradient of an indicator function to align with a solution derived from a (screened) Poisson equation. A crucial requirement of these methods is the input of oriented normals. The Iterative Screened Poisson Reconstruction method[23] introduced an iterative approach to refine the reconstruction process, improving the ability to generate surfaces from point clouds without direct computation of normals. The shape of points [24] introduced a differentiable point-to-mesh layer by employing a differentiable formulation of PSR [1] to generate watertight, topology-agnostic manifold surfaces.

**Neural surface representations.** Recently, the domain of deep learning has spurred significant advances in the implicit neural representation of 3D shapes. Some of these works trained a classifier neural network to construct occupancy fields [9–12] for representing 3D geometries. Poco [12] achieves superior reconstruction performance by introducing convolution into occupancy fields. Ouasfi et al. [25] recently proposed a uncertainty measure method based on margin to learn occupancy fields from sparse point clouds. Compared to occupancy fields, SDFs [2–8] offer a more precise geometric representation by differentiating between interior and exterior spaces through the assignment of signs to distances. Some recent SOTA methods, such as DeepLS [3], using volumetric SDFs to locally learned continuous SDFs, have achieved higher compression, accuracy, and local shape refinement.

**Unsigned distance fields learning.** Although Occupancy fields and SDFs have undergone significant development recently, they are hard to reconstruct surfaces with boundaries or nonmanifold features. G-Shell[26] developed a differentiable shell-based representation for both watertight and non-watertight surfaces. However, UDFs provide a simpler and more natural way to represent general shapes [13–20]. Various methods have been proposed to reconstruct surfaces from point clouds by learning UDFs. CAP-UDF [17] suggested directing 3D query points towards the surface

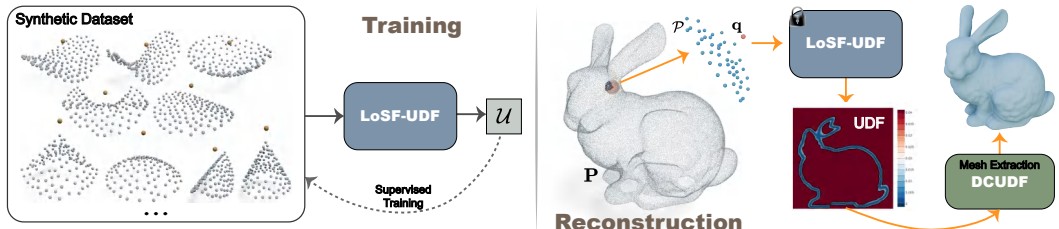

Figure 1: Pipeline. First, we train a UDF prediction network $\mathcal{U}_\Theta$ on a synthetic dataset, which contains a series of local point cloud patches that are independent of specific shapes. Given a global point cloud $\mathbf{P}$, we then extract a local patch $\mathcal{P}$ assigned to each query point $\mathbf{q}$ within a specified radius, and obtain the corresponding UDF values $\mathcal{U}_{\hat{\Theta}}(\mathcal{P}, \mathbf{q})$. Finally, we extract the mesh corresponding to the input point cloud by incorporating the DCUDF[32] framework.

with a consistency constraint to develop UDFs that are aware of consistency. LevelSetUDF [14] learned a smooth zero-level function within UDFs through level set projections. As a supervised approach, GeoUDF [15] estimates UDFs by learning local geometric priors from training on many 3D shapes. DUDF [19] formulated the UDF learning as an Eikonal problem with distinct boundary conditions. UODF [20] proposed unsigned orthogonal distance fields that every point in this field can access to the closest surface points along three orthogonal directions. Instead of reconstructing from point clouds, many recent works [27–30] learn high-quality UDFs from multi-view images for reconstructing non-watertight surfaces. Furthermore, UiDFF [31] presents a 3D diffusion model for UDFs to generate textured 3D shapes with boundaries.

## 3 Method

**Motivation.** Distinct from SDFs, there is no need for UDFs to determine the sign to distinguish between the inside and outside of a shape. Consequently, the UDF values are solely related to the local geometric characteristics of 3D shapes. Furthermore, within a certain radius for a query point, local geometry can be approximated by general mathematical functions. Stemming from these insights, we propose a novel UDF learning framework that focuses on local geometries. We employ local shape functions to construct a series of point cloud patches as our training dataset, which includes common smooth and sharp geometric features. Fig. 1 illustrates the pipeline of our proposed UDF learning framework.

### 3.1 Local shape functions

**Smooth patches.** From the viewpoint of differential geometry [33], the local geometry at a specific point on a regular surface can be approximated by a quadratic surface. Specifically, consider a regular surface $\mathcal{S} : \mathbf{r} = \mathbf{r}(u, v)$ with a point $\mathbf{p}$ on it. At point $\mathbf{p}$, it is possible to identify two principal direction unit vectors, $\mathbf{e}_1$ and $\mathbf{e}_2$, with the corresponding normal $\mathbf{n} = \mathbf{e}_1 \times \mathbf{e}_2$. A suitable parameter system $(u, v)$ can be determined such that $\mathbf{r}_u = \mathbf{e}_1$ and $\mathbf{r}_v = \mathbf{e}_2$, thus obtaining the corresponding first and second fundamental forms as

$$[\mathrm{I}]_{\mathbf{P}} = \begin{bmatrix} E & F \\ F & G \end{bmatrix} = \begin{bmatrix} 1 & 0 \\ 0 & 1 \end{bmatrix}, \quad [\mathrm{II}]_{\mathbf{P}} = \begin{bmatrix} L & M \\ M & N \end{bmatrix} = \begin{bmatrix} \kappa_1 & 0 \\ 0 & \kappa_2 \end{bmatrix}, \tag{1}$$

where $\kappa_1, \kappa_2$ are principal curvatures. Without loss of generality, we assume $\mathbf{p}$ corresponding to $u = v = 0$ and expand the Taylor form at this point as

$$\mathbf{r}(u, v) = \mathbf{r}(0, 0) + \mathbf{r}_u(0, 0)u + \mathbf{r}_v(0, 0)v + \frac{1}{2}[\mathbf{r}_{uu}(0, 0)u^2 +$$
$$\mathbf{r}_{uv}(0, 0)uv + \mathbf{r}_{vv}(0, 0)v^2] + o(u^2 + v^2). \tag{2}$$

Decomposing $\mathbf{r}_{uu}(0, 0), \mathbf{r}_{uv}(0, 0)$, and $\mathbf{r}_{vv}(0, 0)$ along the tangential and normal directions, we can formulate Eq.(2) according to Eq.(1) as

$$\mathbf{r}(u, v) = \mathbf{r}(0, 0) + (u + o(\sqrt{u^2 + v^2}))\mathbf{e}_1 + (v + o(\sqrt{u^2 + v^2}))\mathbf{e}_2$$
$$+ \frac{1}{2}(\kappa_1 u^2 + \kappa_2 v^2 + o(u^2 + v^2)))\mathbf{n} \tag{3}$$

where $o(u^2 + v^2) \approx 0$ is negligible in a small local region. Consequently, by adopting $\{\mathbf{p}, \mathbf{e}_1, \mathbf{e}_2, \mathbf{n}\}$ as the orthogonal coordinate system, we can define the form of the local approximating surface as

$$x = u, \quad y = v, \quad z = \frac{1}{2}(\kappa_1 u^2 + \kappa_2 v^2), \tag{4}$$

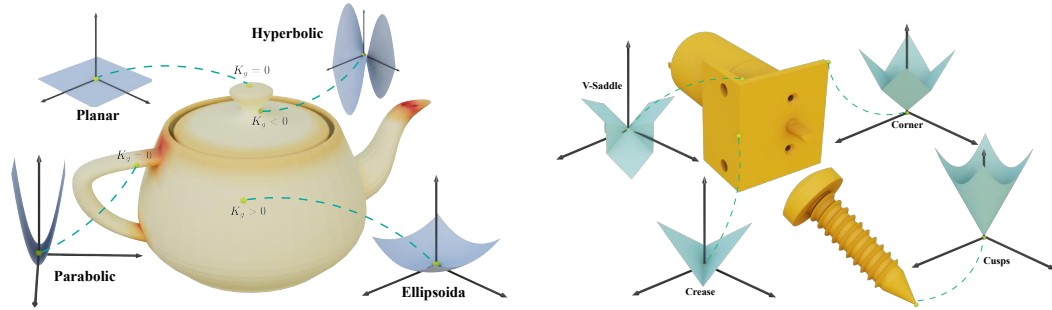

(a) Smooth patches            (b) Sharp patches

Figure 2: Local geometries. (a) For points on a geometry that are differentiable, the local shape at these points can be approximated by quadratic surfaces. (b) For points that are non-differentiable, we can also construct locally approximated surfaces using functions.

which exactly are quadratic surfaces $z = \frac{1}{2}(\kappa_1 x^2 + \kappa_2 y^2)$. Furthermore, in relation to Gaussian curvatures $\kappa_1 \kappa_2$, quadratic surfaces can be categorized into four types: ellipsoidal, hyperbolic, parabolic, and planar. As shown in Fig. 2, for differentiable points on a general geometry, the local shape features can always be described by one of these four types of quadratic surfaces.

**Sharp patches.** For surfaces with sharp features, they are not differentiable at some points and cannot be approximated in the form of a quadratic surface. We categorize commonly seen sharp geometric features into four types, including creases, cusps, corners, and v-saddles, as illustrated in Fig. 2(b). We construct these four types of sharp features in a consistent form $z = f(x, y)$ like smooth patches

$$\text{creases: } z = 1 - h \cdot \frac{|kx - y|}{\sqrt{1 + k^2}}, \text{ cusps: } z = 1 - h \cdot \sqrt{x^2 + y^2},$$
$$\text{corners: } z = 1 - h \cdot \max(|x|, |y|), \text{ v-saddles: } z = 1 - h \cdot |x| + |y| \cdot (\frac{|x|}{x} \cdot \frac{|y|}{y}),$$

(5)

where $h$ can adjust the sharpness of the shape, and $k$ can control the direction of the crease. Fig 3 illustrates various smooth and sharp patches with distinct parameters.

**Synthetic training dataset.** We utilize the mathematical functions introduced above to synthesize a series of point cloud patches for training. As shown in Fig. 3, we first uniformly sample $m$ points $\{(x_i, y_i)\}_{i=1}^m$ within a circle of radius $r_0$ centered at $(0, 0)$ in the $xy$-plane. Then, we substitute the coordinates into Eq.(4-5) to obtain the corresponding $z$-coordinate values, resulting in a patch $\mathcal{P} = \{\mathbf{p}_{i=1}^m\}$, where $\mathbf{p}_i = (x_i, y_i, z(x_i, y_i))$. Subsequently, we randomly collect query points $\{\mathbf{q_i}\}_{i=1}^n$ distributed along the vertical ray intersecting the $xy$-plane at the origin, extending up to a distance of $r_0$. For each query point $\mathbf{q}_i$, we determine its UDF value $\mathcal{U}(\mathbf{q}_i)$, which is either $|\mathbf{q}_i^{(z)}|$ for smooth patches or $1 - |\mathbf{q}_i^{(z)}|$ for sharp patches. Noting that for patches with excessively high curvature or sharpness, the minimum distance of the query points may not be the distance to $(0, 0, z(0, 0))$, we will exclude these patches from our training dataset. Overall, each sample in our synthetic dataset is specifically in the form of $\{\mathbf{q}, \mathcal{P}, \mathcal{U}(\mathbf{q})\}$.

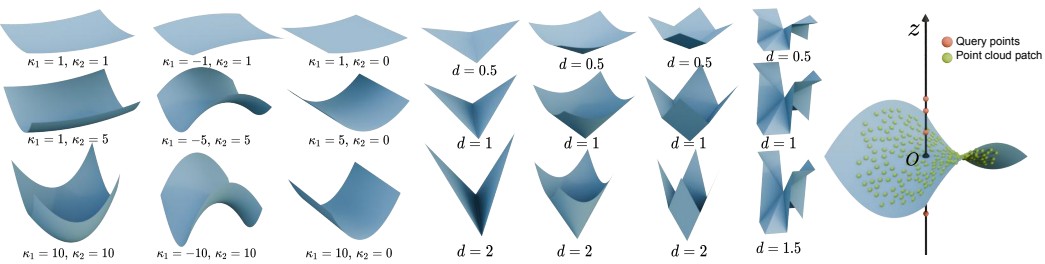

Figure 3: Synthetic dataset for training. By manipulating functional parameters, we can readily create various smooth and sharp surfaces, subsequently acquiring pairs of point cloud patches and query points via sampling.

### 3.2 UDF learning

We perform supervised training on the synthesized dataset which is independent of specific shapes. The network learns the features of local geometries and utilizes an attention-based module to output the corresponding UDF values from the learned features. After training, given any 3D point clouds and a query point in space, we extract the local point cloud patch near the query, which has the same form as the data in the training dataset. Consequently, our network can predict the UDF value at that query point based on this local point cloud patch.

#### 3.2.1 Network architecture

For a sample $\{\mathbf{q}, \mathcal{P} = \{\mathbf{p}_i\}_{i=1}^m, \mathcal{U}(\mathbf{q})\}$, we first obtain a latent code $\mathbf{f}_p \in \mathbb{R}^{l_p}$ related to the local point cloud patch $\mathcal{P}$ through a Point-Net [34] $\mathcal{F}_p$. To derive features related to distance, we use relative vectors from the patch points to the query point, $\mathcal{V} = \{\mathbf{p}_i - \mathbf{q}\}_{i=1}^m$, as input to a Vectors-Net $\mathcal{F}_v$, which is similar to the Point-Net $\mathcal{F}_p$. This process results in an additional latent code $\mathbf{f}_v \in \mathbb{R}^{l_v}$. Subsequently, we apply a cross-attention module [35] to obtain the feature codes for the local geometry,

$$\mathbf{f}_G = \text{CrossAttn}(\mathbf{f}_p, \mathbf{f}_v) \in \mathbb{R}^{l_G}, \tag{6}$$

where we take $\mathbf{f}_p$ as the Key-Value (KV) pair and $\mathbf{f}_v$ as the Query (Q). In our experiments, we set $l_p = l_v = 64$, and $l_G = 128$. Based on the learned geometric features, we aim to fit the UDF values from the distance within the local point cloud. Therefore, we concatenate the distances $\mathbf{d} \in \mathbb{R}^m$ induced from $\mathcal{V}$ with the latent code $\mathbf{f}_G$, followed by a series of fully connected layers to output the predicted UDF values $\mathcal{U}_\Theta(\mathbf{q})$. Fig. 4 illustrates the overall network architecture and data flow.

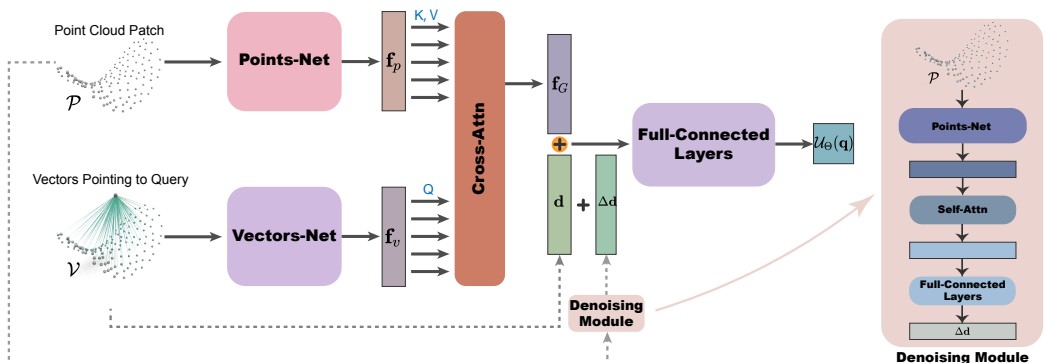

Figure 4: Network architecture of LoSF-UDF.

**Denoising module.** In our network, even if point cloud patches are subjected to a certain degree of noise or outliers, their representations in the feature space should remain similar. However, distances induced directly from noisy vectors $\mathcal{V}$ will inevitably contain errors, which can affect the accurate prediction of UDF values. To mitigate this impact, we introduce a denoising module that predicts displacements $\Delta \mathbf{d}$ from local point cloud patches, as shown in Fig. 4. We then add the displacements $\Delta \mathbf{d}$ to the distances $\mathbf{d}$ to improve the accuracy of the UDF estimation.

#### 3.2.2 Training and evaluation

**Data augmentation.** During the training process, we scale all pairs of local patches $\mathcal{P}$ and query points $\mathbf{q}$ to conform to the bounding box constraints of $[-0.5, 0.5]$, and the corresponding GT UDF values $\mathcal{U}(\mathbf{q})$ are scaled by equivalent magnitudes. Given the uncertain orientation of local patches extracted from a specified global point cloud, we have applied data augmentation via random rotations to the training dataset. Furthermore, to enhance generalization to open surfaces with boundaries, we randomly truncate 20% of the smooth patches to simulate boundary cases. To address the issue of noise handling, we introduce Gaussian noise $\mathcal{N}(0, 0.1)$ to 30% of the data in each batch during every training epoch.

**Loss functions.** We employ $L_1$ loss $\mathcal{L}_u$ to measure the discrepancy between the predicted UDF values and the GT UDF values. Moreover, for the displacements $\Delta \mathbf{d}$ output by the denoising module,

we employ $L_1$ regularization to encourage sparsity. Consequently, we train the network driven by the following loss function,

$$\mathcal{L} = \mathcal{L}_u + \lambda_d \mathcal{L}_r, \quad \text{where } \mathcal{L}_u = |\mathcal{U}(\mathbf{q}) - \mathcal{U}_\Theta(\mathbf{q})|, \ \mathcal{L}_r = |\Delta \mathbf{d}|, \tag{7}$$

where we set $\lambda_d = 0.01$ in our experiments.

**Evaluation.** Given a 3D point cloud $\mathbf{P}$ for reconstruction, we first normalize it to fit within a bounding box with dimensions ranging from $[-0.5, 0.5]$. Subsequently, within the bounding box space, we uniformly sample grid points at a specified resolution to serve as query points. Finally, we extract the local geometry $\mathcal{P}_\mathbf{p}$ for each query point by collecting points from the point cloud that lie within a sphere of a specified radius centered on the query point. We can obtain the predicted UDF values by the trained network $\mathcal{U}_{\Theta^*}(\mathbf{q}, \mathcal{P}_\mathbf{q})$, where $\Theta^*$ represents the optimized network parameters. Note that for patches $\mathcal{P}_\mathbf{p}$ with fewer than 5 points, we set the UDF values as a large constant. Finally, we extract meshes from the UDFs using the DCUDF model [32].

# 4 Experiments

## 4.1 Experiment setup

**Datasets.** To compare our method with other state-of-the-art UDF learning approaches, we tested it on various datasets that include general artificial objects from the field of computer graphic. Following previous works [30, 17, 14], we select the "Car" category from ShapeNet[21], which has a rich collection of multi-layered and non-closed shapes. Furthermore, we select the real-world dataset DeepFashion3D[36] for open surfaces, and ScanNet[37] for large outdoor scenes. To assess our model's performance on actual noisy inputs, we conducted tests on real range scan dataset [38] following the previous works[17, 14].

**Baselines.** For our validation datasets, we compared our method against the state-of-the-art UDF learning models, which include unsupervised methods like CAP-UDF[17], LevelSetUDF[14], and DUDF[19], as well as the supervised learning method, GeoUDF[15]. We trained GeoUDF independently on different datasets to achieve optimal performance. Table. 1 shows the qualitative comparison between our methods and baselines. To evaluate performance, we calculate the Chamfer Distance (CD) and F1-Score (setting thresholds of 0.005 and 0.01) metrics between the ground truth meshes and the meshes extracted from the UDFs out by our model and each baseline model. For a fair comparison, we test all baseline models using the DCUDF[32] method. All experimental procedures are executed on NVIDIA RTX 4090 and A100 GPUs.

| Methods | Input | Normal | Learning Type | Feature Type | Noise | Outlier |
|---|---|---|---|---|---|---|
| CAP-UDF [17] | Dense | Not required | Unsupervised | Global | ✗ | ✗ |
| LevelSetUDF [14] | Dense | Not required | Unsupervised | Global | ✓ | ✗ |
| GeoUDF [15] | Sparse | Not required | Supervised | Local | ✗ | ✗ |
| DUDF [19] | Dense | Required | Unsupervised | Global | ✗ | ✗ |
| Ours | Dense | Not required | Supervised | Local | ✓ | ✓ |

Table 1: Qualitative comparison of different UDF learning methods. "Normal" indicates whether the method requires point cloud normals during learning. "Feature Type"' refers to whether the information required during training is global or local. "Noise" and "Outlier" indicate whether the method can handle the presence of noise and outliers in point clouds.

## 4.2 Experimental results

**Synthetic data.** For general 3D graphic models, ShapeNetCars, and Deep-Fashion3D, we obtain dense point clouds by randomly samping on meshes. Considering that GeoUDF [15] is a supervised method, we retrain it on ShapeNetCars, and DeepFashion3D, which are randomly partitioned into training (70%), testing (20%), and validation subsets (10%). All models are evaluated in the validation sets, which remain unseen by any of the UDF learning models prior to evaluation. The first three rows of Fig. 5 show the visual comparison of reconstruction results, while Tab. 2 presents the quantitative comparison results of CD and F1-score. We

CAP-UDF    CAP-UDF+DCUDF    GeoUDF    GeoUDF+DCUDF

test each method using their own mesh extraction technique, as shown in the inset figure, which display obvious visual artifacts such as small holes and non-smoothness. We thus apply DCUDF [32] , the state-of-art method, to each baseline model , extracting the surfaces as significantly higher quality meshes. Since our method utilizes DCUDF for surface extraction, we adopt it as the default technique to ensure consistency and fairness in comparisons with the baselines. Our method achieves stable results in reconstructing various types of surfaces, including both open and closed surfaces, and exhibits performance comparable to that of the SOTA methods. Noting that DUDF[19] requires normals during training, and GeoUDF utilizes the KNN approach to determine the nearest neighbors of the query points. Although DUDF and GeoUDF achieve better evaluations, they are less stable when dealing with point clouds with noise and outliers.

| | method | Clean CD↓ | F1↑ $F1^{0.005}$ | $F1^{0.01}$ | Noise CD↓ | F1↑ $F1^{0.005}$ | $F1^{0.01}$ | Outlier CD↓ | F1↑ $F1^{0.005}$ | $F1^{0.01}$ |
|---|---|---|---|---|---|---|---|---|---|---|
| ShapeNetCars [21] | CAP-UDF [17] | 2.432 | 0.523 | 0.888 | 2.602 | 0.194 | 0.381 | 4.982 | 0.183 | 0.314 |
| | LevelSetUDF [14] | 1.534 | 0.561 | 0.908 | 2.490 | 0.209 | 0.401 | 4.177 | 0.199 | 0.363 |
| | GeoUDF [15] | 1.257 | 0.571 | 0.889 | 1.232 | 0.351 | 0.873 | 4.870 | 0.187 | 0.346 |
| | DUDF [19] | 0.568 | 0.903 | 0.991 | 3.180 | 0.312 | 0.527 | 4.235 | 0.168 | 0.308 |
| | **Ours** | 1.085 | 0.510 | 0.938 | 1.114 | 0.427 | 0.922 | 1.272 | 0.485 | 0.771 |
| DeepFashion3D [36] | CAP-UDF [17] | 1.660 | 0.417 | 0.818 | 1.892 | 0.336 | 0.542 | 4.941 | 0.172 | 0.430 |
| | LevelSetUDF [14] | 1.500 | 0.403 | 0.856 | 1.488 | 0.453 | 0.729 | 4.328 | 0.203 | 0.468 |
| | GeoUDF [15] | 0.652 | 0.864 | 0.977 | 1.258 | 0.380 | 0.957 | 4.463 | 0.147 | 0.300 |
| | DUDF [19] | 0.381 | 0.991 | 0.998 | 1.894 | 0.334 | 0.535 | 4.970 | 0.144 | 0.272 |
| | **Ours** | 0.932 | 0.652 | 0.983 | 1.150 | 0.361 | 0.976 | 1.029 | 0.549 | 0.973 |

Table 2: Quantitative evaluation of UDF learning methods (CD score is multiplied by 100).

**Noise & outliers.** To evaluate our model with noisy inputs, we added Gaussian noise $\mathcal{N}(0, 0.0025)$ to the clean data across all datasets for testing. The middle three rows in Fig. 5 display the reconstructed surface results from noisy point clouds, and Tab. 2 also presents the quantitative comparisons. It can be observed that our method can robustly reconstruct smooth surfaces from noisy point clouds. Additionally, we tested our method's performance with outliers by converting 10% of the clean point cloud into outliers, as shown in the last three rows of Fig. 5. To further demonstrate the robustness of our method, we conducted experiments on point clouds with higher percentage of outliers. Our framework is able of reconstructing reasonable surfaces even with 50% outliers. We also tested the task on point clouds containing both noise and outliers. Please refer to Fig. 9 in the Appendix for the corresponding results.

**Real-world scanned data.** Dataset [38] provide several real-world scanned point clouds, as illustrated in Fig. 6 (Left), we evaluate our model on the dataset to demonstrate the effectiveness. Our approach can reconstruct smooth surfaces from scanned data containing noise and outliers. However, our model cannot address the issue of missing parts. This limitation is due to the local geometric training strategy, which is independent of the global shape. Additionally, we conduct tests on large scanned scenes to evaluate our algorithm, as shown in Fig. 6 (Right).

## 4.3 Analysis & ablation studies

**Efficiency**. As a supervised UDF learning method, our approach has a significant improvement in training efficiency compared to GeoUDF [15]. As shown in the insert table, we calculate the data storage space required by GeoUDF when using ShapeNet as a training dataset. This includes the GT UDF values and point cloud data needed during the training process. Our

| Method | Storage (GB) | Data-prep (min) | Training (h) |
|---|---|---|---|
| GeoUDF | 120 | 0.5 | 36 |
| **Ours** | **0.59** | **0.02** | **14.5** |

synthetic point cloud patches training dataset occupies under 1GB, which is merely 0.5% of the storage needed for GeoUDF. Our network is very lightweight, with only 653KB of trainable parameters and a total parameter size of just 2MB. Additionally, we highlight time-saving benefits. The provided table illustrates the duration required to produce a single data sample for dataset preparation ("Data-prep"), as well as the total time for training ("Training").

**Patch radius.** During the evaluation phase, the radius $r$ used to find the nearest points for each query point determines the size of the extracted patch and the range of effective query points in the space. As shown in Fig. 7, we analyzed the impact of different radii on the reconstruction results. An excessively small $r$ will generate artifacts, while an overly large $r$ will lose many details. In our experiments, we generally set $r$ to 0.018.

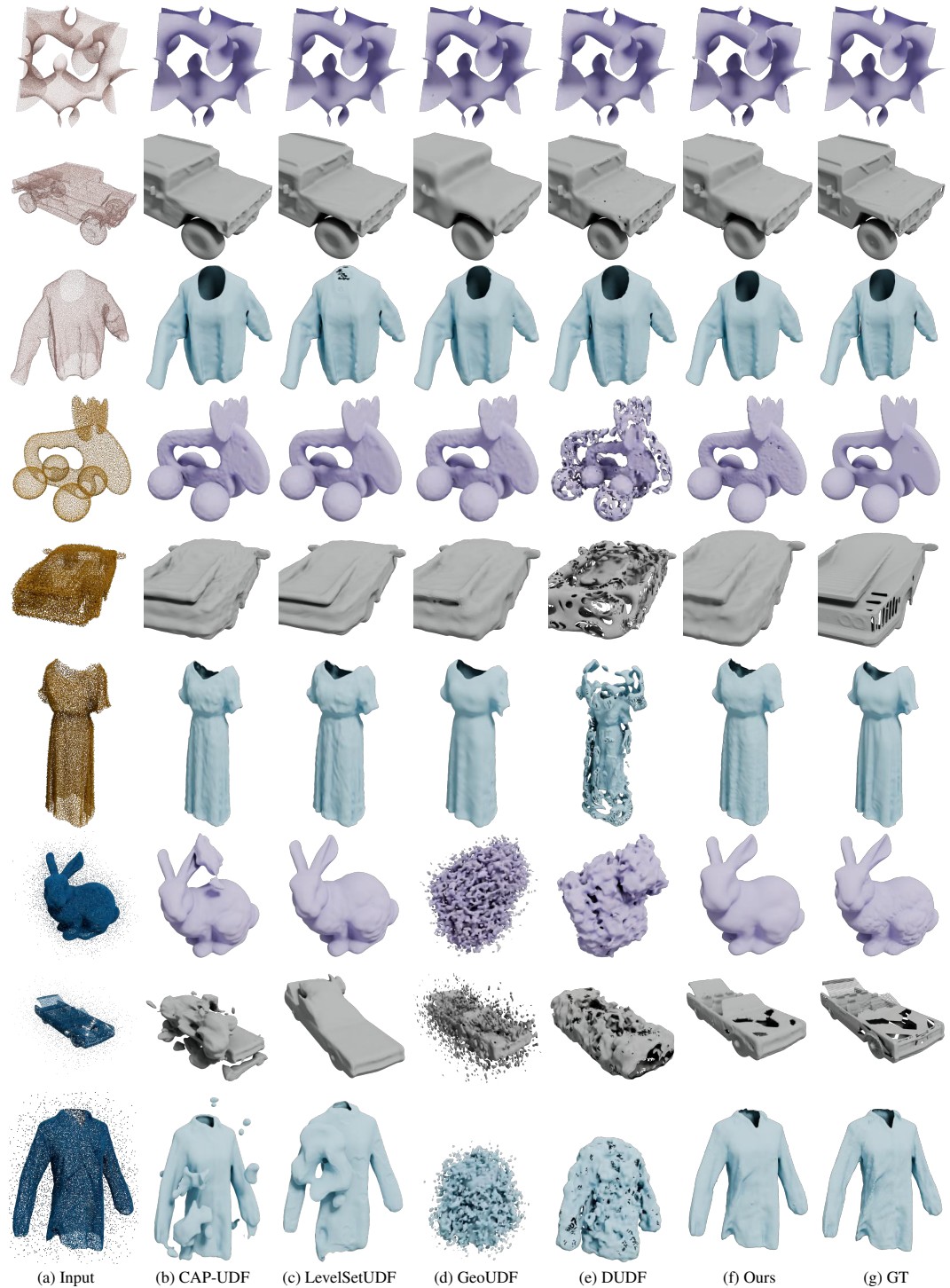

| (a) Input | (b) CAP-UDF | (c) LevelSetUDF | (d) GeoUDF | (e) DUDF | (f) Ours | (g) GT |

Figure 5: Visual comparisons on the synthetic dataset. First three rows: uniformly sampled points. Meddle three rows: point clouds with 0.25% added noise. Last three rows: point clouds with 10% outliers. All point clouds here have 48K points, except for the Bunny model, which has 100K points. We refer readers to the appendix for more visual results.

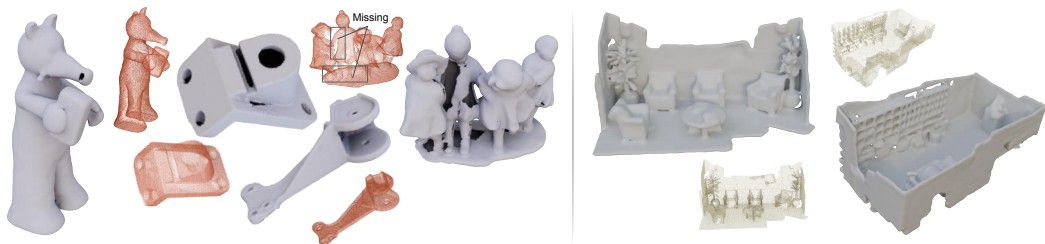

Figure 6: Reconstructed surfaces from real-world scanned point clouds.



| GT | $r = 0.08$ | $r = 0.10$ | $r = 0.20$ | $r = 0.30$ |

Figure 7: Comparison of different radii for extracting patches from the point cloud on reconstruction results.

**Denoising module.** Our framework incorporates a denoising module to handle noisy point clouds. We conducted ablation experiments to verify the significance of this module. Specifically, we set $\lambda_d = 0$ in the loss function Eq. (7) to disable the denoising module, and then retrained the network. As illustrated in Fig. 8, we present the reconstructed surfaces for the same set of noisy point clouds with and without the denosing module, respectively.

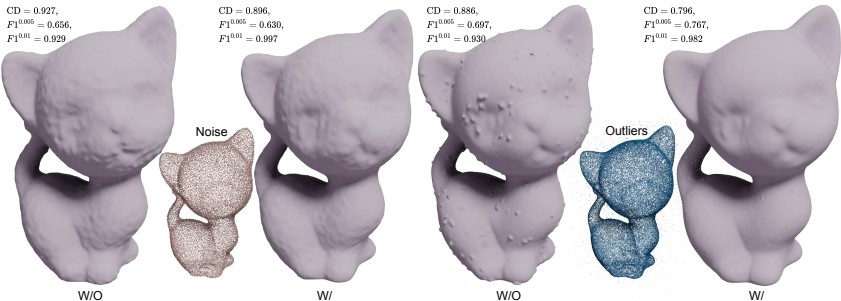

Figure 8: Ablation on denoising module: Reconstructed surfaces from the same point clouds with noise/outliers corresponding to framework with and without the denoising module, respectively.

## 5 Conclusion

In this paper, we introduce a novel and efficient neural framework for surface reconstruction from 3D point clouds by learning UDFs from local shape functions. Our key insight is that 3D shapes exhibit simple patterns within localized regions, which can be exploited to create a training dataset of point cloud patches represented by mathematical functions. As a result, our method enables efficient and robust surfaces reconstructions without the need for shape-specific training. Extensive experiments on various datasets have demonstrated the efficacy of our method. Moreover, our framework achieves superior performance on point clouds with noise and outliers.

**Limitations & future work.** Owing to its dependence solely on local geometric features, our approach fails to address tasks involving incomplete point cloud reconstructions. However, as a lightweight framework, our model can readily be integrated into other unsupervised methods to combine the global features with our learned local priors. Furthermore, in our future work, we intend to design a method that dynamically adjusts the radius based on local feature sizes [39] of 3D shapes when extracting local point cloud patches for queries, aiming to improve the accuracy of the reconstruction.

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

# A  Appendix

## A.1  Network details

The two PointNets used in our network to extract features from point cloud patches $\mathcal{P}$ and vectors $\mathcal{V}$ consist of four ResNet blocks. In addition, the two fully connected layer modules in our framework consist of three layers each. To ensure non-negativity of the UDF values output by the network, we employ the softplus activation function.

## A.2  Robustness to outliers

Our method can reconstruct relatively accurate geometry from point clouds with 10% added outliers and reasonably smooth surfaces from point clouds with even higher outlier ratios. Furthermore, our approach can reconstruct high-quality geometry from point clouds containing both noise and outliers, as shown in Fig. 9.

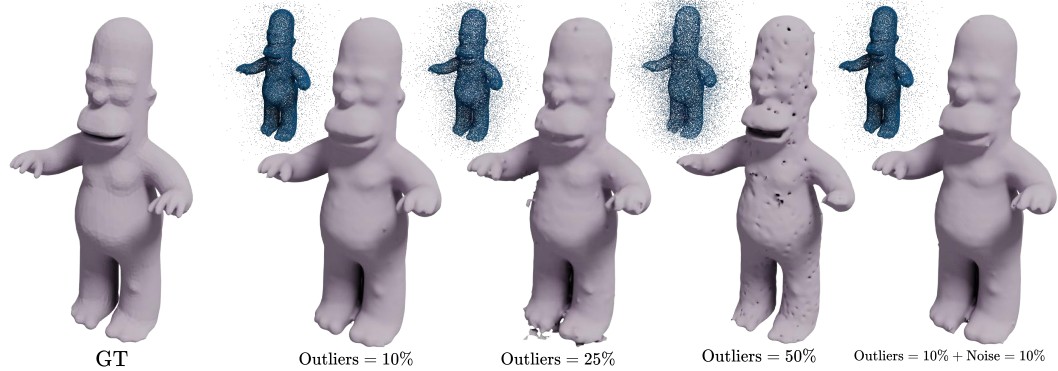

GT      Outliers = 10%      Outliers = 25%      Outliers = 50%      Outliers = 10% + Noise = 10%

Figure 9: Our model demonstrates robustness to more outliers.

## A.3  More results

As shown in Fig. 10 and Fig. 11, we provide more visual comparisons on the DeepFashion3D and ShapeNetCars dataset, using point clouds containing noise and outliers.

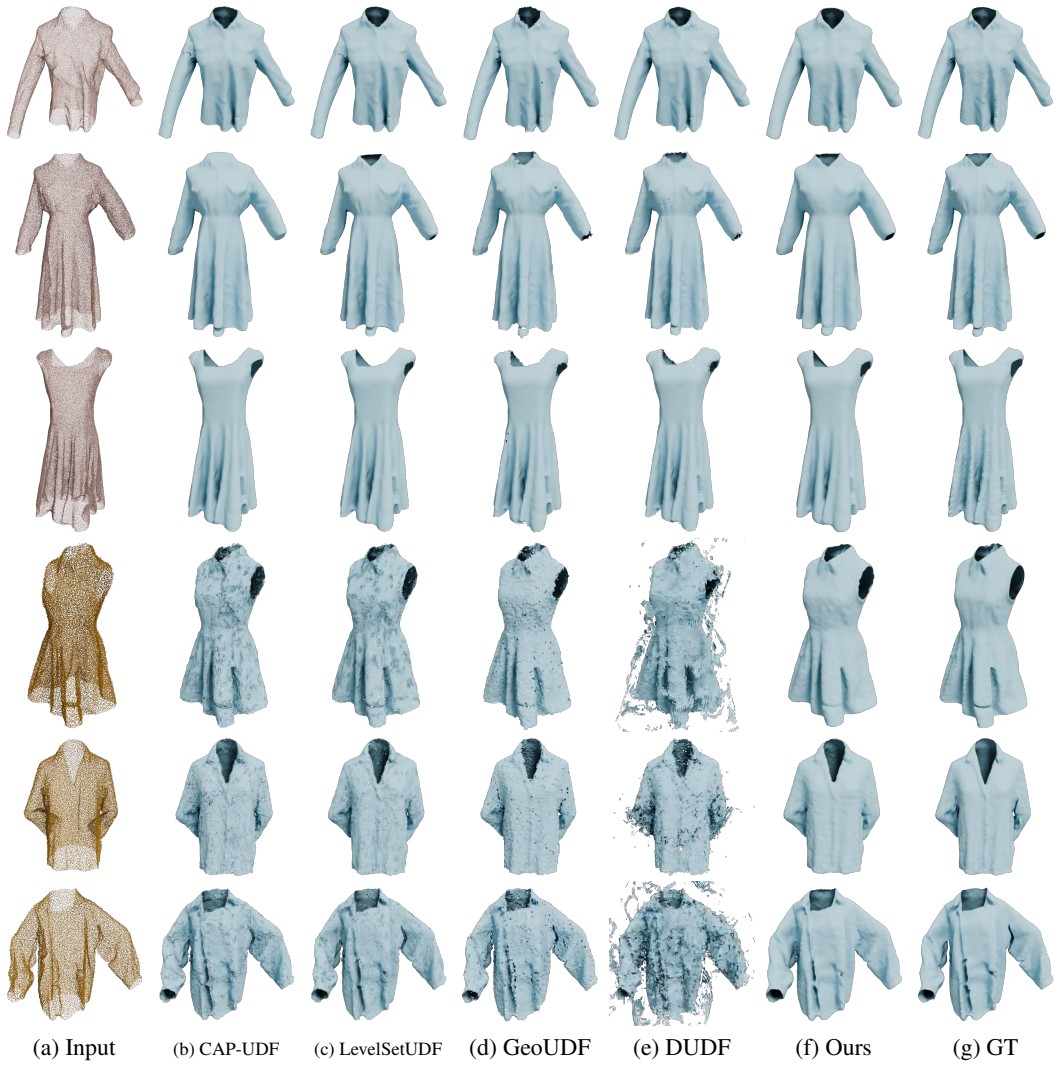

|     |     |     |     |     |     |     |
|-----|-----|-----|-----|-----|-----|-----|
| (a) Input | (b) CAP-UDF | (c) LevelSetUDF | (d) GeoUDF | (e) DUDF | (f) Ours | (g) GT |

Figure 10: More visual results on the DeepFashion3D dataset. Top three rows: Reconstruction results under noise-free conditions. Bottom three rows: Reconstruction results under noise condition.

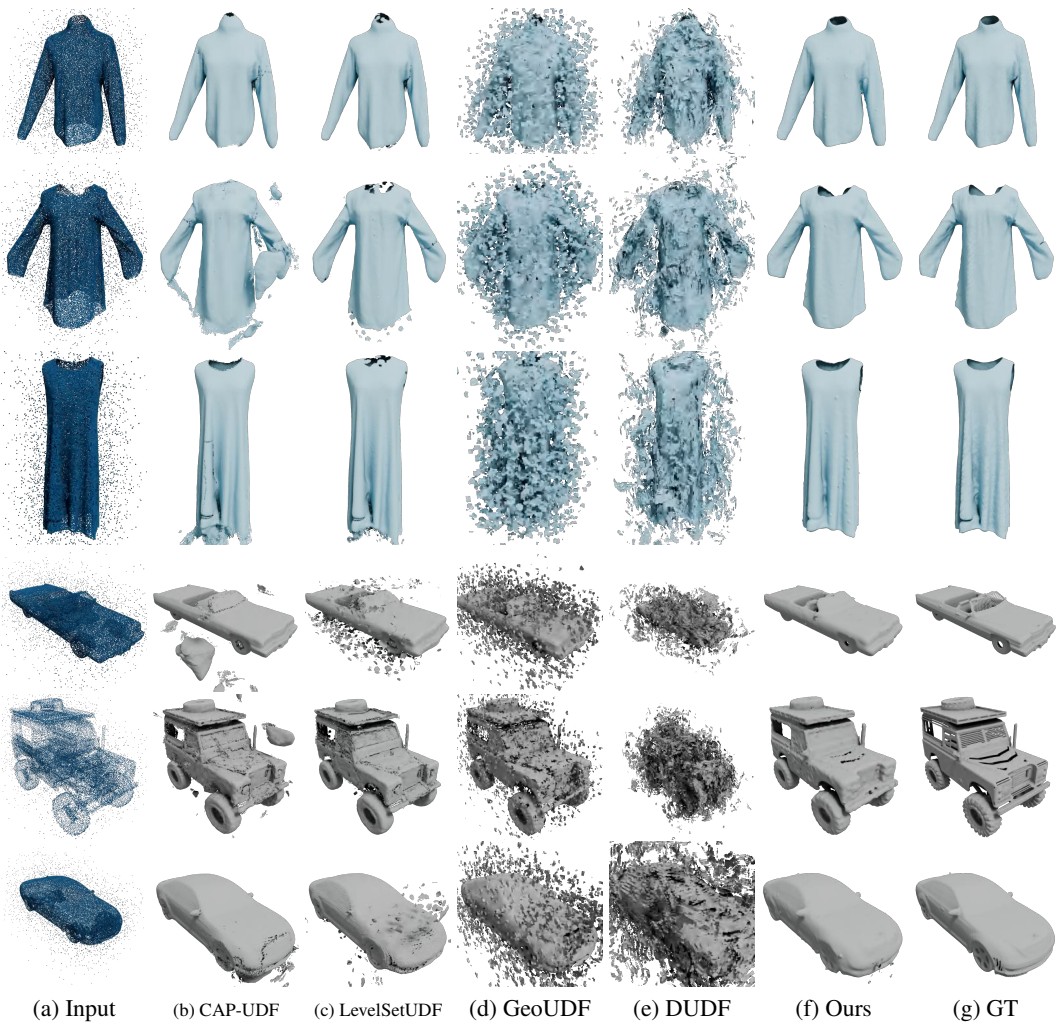

| (a) Input | (b) CAP-UDF | (c) LevelSetUDF | (d) GeoUDF | (e) DUDF | (f) Ours | (g) GT |

Figure 11: More visual results on the synthetic datasets with outliers.

