# OpenReview forum: "Learning Unsigned Distance Fields from Local Shape Functions for 3D Surface Reconstruction"
_NeurIPS.cc/2024/Conference — Submitted to NeurIPS 2024_

### Official Review · Reviewer_1t3W · 2024-07-09

**Soundness:** 3
**Presentation:** 3
**Contribution:** 2
**Rating:** 5
**Confidence:** 5

**Summary:**

This paper introduces a new method to learning unsigned distance functions. Specifically, the method leverages local shape priors which brings geometry priors and is also able to handle noises and outliers. The results demonstrate that the proposed method outperforms previous baselines, especially in the corrupted situations.

**Strengths:**

1. The idea of learning local shape functions is widely explored in the SDF or occupancy based methods, but is not yet introduced to the field of UDF-based reconstruction. This paper combines the strengths of local functions and UDF which can represent shapes with arbitrary typologies.
2. The paper is overall easy to follow.
3. The comparison are comprehensive by introducing recent works as the baselines.
4. The method can better handle the noises and outliers compared to the previous prior-free methods.

**Weaknesses:**

1. The idea is straightforward to learn local shape functions as the prior for global shape reconstruction. Similar ideas are proposed in SDF-based methods. I would like to see more insight in the differences to the previous methods and also the importance of introducing the local geometric priors for UDF learning.
Local Implicit Grid Representations for 3D Scenes (CVPR 2020)
Surface Reconstruction from Point Clouds by Learning Predictive Context Priors (CVPR 2022)
Deep local shapes: Learning local sdf priors for detailed 3d reconstruction (ECCV 2020)
2. The visualization is not quite convincing in the geometry details. For example, the reconstructions of real-scans in Fig.6 seems worse than the results of other methods in their papers. The local shape functions is expected to produce reconstruction with more details and sharper edges, but the scene reconstructions lack details.
3. Also, some reconstructions are too fat, which in my opinion, is caused by the inaccurate UDF near the zero-level set, since the method use DCUDF for UDF meshing.

**Questions:**

More references on the local shape functioning and UDF learning will be helpful.

Local Implicit Grid Representations for 3D Scenes (CVPR 2020)
CAP-UDF: Learning Unsigned Distance Functions Progressively from Raw Point Clouds with Consistency-Aware Field Optimization (TPAMI 2024)

Can the proposed method handle large scale scenes like KITTI?

**Limitations:**

Please refer to the strengths and weaknesses above.

---

> ### Author Rebuttal · Authors · 2024-08-07
>
> Thank you for recognizing the contributions of our paper. We also appreciate your valuable suggestions.
> In the following, we address your main concerns. Please refer to the supplementary one-page PDF for figures and tables.
>
> **Q1: The visualization is not quite convincing in the geometry details.**
> We reiterate that our method introduces a novel strategy for learning UDFs from local shape functions. It requires only a single training session on a synthetic dataset composed of local patches represented by simple mathematical functions, and it demonstrates strong generalization capabilities across various point cloud shapes. Our method not only achieves comparable or superior performance to SOTA methods but does so at a significantly lower  training cost. Its independence from specific shape categories greatly enhances its applicability. For models with rich geometric details, our method cannot reconstruct them well due to feature bias from our synthetic dataset. Moreover, we provide a strategy combining our method with unsupervised methods to enhance geometric details. Specifically, leveraging its efficiency and generalization capabilities, our method can serve as an effective initialization for training network. The initialization provided by our method not only stabilizes the network but also accelerates its convergence, cutting the training time from 30 minutes to just 10 minutes. Fig.5 showcases three examples where this strategy significant enhances geometric accuracy and fidelity.
>
> **Q2: More references on the local shape functions and UDF learning.**
> Thank you for your suggestion. We acknowledge that the paper "Local Implicit Grid Representations for 3D Scenes" (CVPR 2020) introduces a new 3D shape representation called Local Implicit Grid Representations, which is indeed related to our work. The work "CAP-UDF" (TPAMI 2024) is an improvement over the baseline method compared in our paper. We will conduct a more detailed survey and discussion on UDF learning and representations based on local shape functions.
>
> **Q3: Tests on KITTI dataset.**
> Thanks for your suggestion, and we tested our method on point clouds of the KITTI dataset. Due to the overly sparse point cloud density (refer to Fig.4 for point density analysis), the reconstruction failed. However, other baseline methods did not perform well on KITTI dataset either.

---

> > ### Comment · Reviewer_1t3W · 2024-08-13
> > **Response to authors**
> >
> > Thanks for the response. I will maintain my original score as borderline accept.

---

### Official Review · Reviewer_15wp · 2024-07-09

**Soundness:** 1
**Presentation:** 2
**Contribution:** 3
**Rating:** 4
**Confidence:** 3

**Summary:**

The paper proposes an approach to reconstruct 3D surfaces from point clouds, using unsigned distance fields. The proposed approach consists in training a specific neural network architecture to predict UDF values from local point cloud patches, which can be triangulated using UDF meshing methods. The paper mathematically analyses the possible local patches that can appear, smooth and sharp, and trains the proposed architecture with them. Once trained, the neural network is queried for UDF values from a point cloud, and it computes them by extracting a local point cloud patch for each query point and applying the information learned during training. A denoising module is employed to reduce the impact of noise and outliers. Experiments are carried out on ShapeNet cars and DeepFashion3D, synthetically sampling point clouds from the dataset meshes and reconstructing surfaces using a number of baselines, an extracting meshes using DCUDF. Results show that the performance of the proposed method is not the strongest on clean data, but it is so in the presence of noise and/or outliers. Experiments on real-world point clouds are shown qualitatively.

**Strengths:**

The analysis of the possible kinds of local surface patches is well thought out and significative, enabling the possibility to treat the problem of surface reconstruction locally instead of globally, with an extensive training set that covers virtually all possible cases. I believe this is a good contribution, and the main strength in the paper.
The performance of the method on noisy data, albeit the noise was introduced synthetically, is good; the performance with synthetic outliers looks impressive.
The writing quality is generally good, with some exceptions on clarity (see weaknesses).

**Weaknesses:**

I believe there are a few weaknesses in the paper, mostly regarding clarity, architecture validation and experiments. I list them in no particular order.
1) The paper proposes a complex architecture, consisting of two branches, a cross-attention module, fully connected layers and a denoising module. The latter is only qualitatively justified in Fig.8, but the rest of the architecture is not validated in any way. A (quantitative) experimental evaluation of why this architecture is suitable for the task would be needed, in my opinion, as well as the intuition that led the researchers to arrive to this final architecture.
2) The cross-attention mechanism is unclear to me: the two branches (points and vectors net) output a latent code. What is the meaning of a cross-attention module on a single input token (actually one as K-V, one as Q)?
3) In the experiments, a few details are not specified, for example the query grid resolution and the number of points used for the Chamfer distance.
4) A time evaluation is completely missing. The paper claims "Our method is computationally efficient" in the introduction, which is partially justified by the comparatively better training times and storage requirements with respect to GeoUDF, but there is no evaluation of the time required by the method to reconstruct a surface, compared to the other methods. Notice also that DCUDF is an extremely slow meshing algorithm, so the evaluation should be performed both with it and without it.
5) The paper claims "superior performance in surface reconstruction from both synthetic point clouds and real scans, even in the presence of noise and outliers". The experiments however show no superiority in the absence of noise and outliers. Moreover, there are no quantitative experiments on real scans, making it impossible to evaluate whether the noise robustness on synthetic noise also translates to real data. Additionally, the qualitative experiments shown on real data lack comparison with the baselines. Thus, in general, I find the experimental section incomplete and not fully convincing.

**Questions:**

Relating to some of the weaknesses, I would ask the authors:
1) Could you clarify the role and mechanism of the cross-attention module in the architecture?
2) Could you specify the missing details? (grid resolution, number of points for Chamfer distance)
3) If you already have them in the logs or if time allows, could you provide the timings for your methods and the baselines?
4) Why are surfaces from real data not evaluated quantitatively?
5) In the conclusions, addressing the limitation of the proposed method on incomplete point clouds, you mention the possibility to readily integrate your framework with other methods, but it is unclear to me how this could be possible. Could you provide a brief explanation?

**Limitations:**

A limitation of the method on completing incomplete surfaces has been correctly addressed and disclosed.
The limited performance on clean data is shown in the experiments, but not acknowledged in the claims in the introduction.
A quantitative assessment of the performance on real scans and of the time required to reconstruct the surface are missing, making it harder to fully evaluate the possible limitations of the method with respect to the claims.

---

> ### Author Rebuttal · Authors · 2024-08-07
>
> Thank you for recognizing the contributions of our paper. We also appreciate your valuable suggestions.
> In the following, we address your main concerns. Please refer to the supplementary one-page PDF for figures and tables.
>
> **Q1: The intuition to design the network architecture and more ablation studies.**
> Our intuition to design the network architecture: Our main goal is to derive the UDF value for a query point by learning the local geometry within a radius $r$. To achieve this, we utilize Points-Net to capture the point cloud features $\mathbf{f}_p$ of local patches. This process enables the local geometry extracted from test data to align with the synthetic data through feature matching, even in the presence of noise or outliers. Vectors-Net is tasked with learning the features $\mathbf{f}_v$ of the set of vectors pointing towards the query point, which includes not only the position of the query point but also its distance information. The Cross-Attn module then processes these local patch features $\mathbf{f}_p$ as keys and values to query the vector features $\mathbf{f}_v$, which contain distance information, returning the most relevant feature $\mathbf{f}_G$ that determines the UDF value.
>
> To validate the effectiveness of each component, we conducted the following ablation studies: (a) removing the Points-Net and Cross-Attn modules; (b) removing the Cross-Attn module. We tested the performance on noisy point clouds. The results in Figure 3 demonstrate the important role of each component.
>
> **Q2: Details about the experiments.**
> We apologize for the omission of these details. The sampling resolution for query points and the mesh extraction grid resolution were both set to 512. When computing the Chamfer Distance (CD), we sampled 100,000 points each from the gt mesh and the reconstructed mesh.
>
> **Q3:Time evaluation.**
> We have supplemented the runtime information in Table.1. All tests were conducted using an Intel i9-13900K CPU and an NVIDIA RTX 4090 GPU-24GB. As supervised methods, CAPUDF, LevelsetUDF, and DUDF require extensive UDF learning time. Although their mesh extraction process is relatively fast, the resulting mesh quality is somewhat poor and necessitates normal.
> Comparative results show that supervised, local feature-based methods like GeoUDF and ours outperform unsupervised methods in computational efficiency.
> However, GeoUDF's UDF inference relies on the training dataset's category.
> The evaluation in our manuscript (Sec.4.3) also demonstrates that our method surpasses GeoUDF in terms of generalization, storage complexity, and training time.
>
> **Q4: Quantitative evaluation on real scan data.**
> We show the visual comparison and the quantitative evaluations of different methods on real scan dataset in Fig.2.
>
> **Q5: A brief explanation about integrating LoSF-UDF with other unsupervised methods.**
> Thanks for your interest in this point. We have already conducted some preliminary experiments. Due to the high efficiency and generalizability of our method, it serves as an effective initialization for unsupervised methods. We train a Siren network (Sitzmann et al. NeurIPS, 2020) using UDF values derived from our pre-trained LoSF-UDF for supervision, employing MSE, Eikonal and normal alignment losses. The initialization provided by our method not only stabilizes the network but also accelerates its convergence, cutting the training time from 30 minutes to just 10 minutes. Fig.5 showcases three examples where this strategy significant enhances geometric accuracy and fidelity.

---

> > ### Comment · Reviewer_15wp · 2024-08-10
> >
> > Thank you for your responses.
> >
> > I still do no understand two things:
> >
> > 1) The role of cross attention: my understanding was that the points-net extracts one feature per patch and so does the vectors-net. In this case, as I mentioned in my review, I do not understand the role of cross attention. In your comment you seem to suggest that the points-net extract more than one feature. Is that so?
> >
> > 2) When the Siren network is used to reconstruct the surface, it achieves better results than the proposed method. Is this so even for a vanilla Siren network (i.e. without your initialization)? It should be included in the baselines in the paper.

---

> > > ### Author Response · Authors · 2024-08-11
> > >
> > > Thank you for your questions.
> > >
> > > **Q: The role of cross attention.**
> > > In our network, the outputs of Points-Net and Vectors-Net are indeed one-dimensional vectors, specifically $\mathbf{f}_p\in\mathbb{R}^{64}$ and $\mathbf{f}_v\in\mathbb{R}^{64}$ in our experiments. The Cross-Attn module plays a crucial role in effectively fusing these two feature sets.
> > > We begin by detailing how the Cross-Attn module processes these vectors. We obtain the query $Q$, key $K$, and value $V$ through the following linear transformations:
> > > $$Q=\mathbf{W}_q\cdot\mathbf{f}_v,\,\,K=\mathbf{W}_k\cdot\mathbf{f}_q,\,\,V=\mathbf{W}_v\cdot\mathbf{f}_q,$$
> > > where $\mathbf{W}_q, \mathbf{W}_k, \mathbf{W}_v\in\mathbb{R}^{64\times64}$ are learnable weight matrices. Then, we compute attention scores as
> > > $$S=\text{Softmax}(\frac{Q K^T}{\sqrt{d}}),$$
> > > where $d=64$ is the dimension. The attention scores quantify the correlation between $\mathbf{f}_p$ and $\mathbf{f}_v$, with $\mathbf{f}_p$ encapsulating local patch features and $\mathbf{f}_v$ providing distance information. The final feature vector $\mathbf{f}_G$ is obtained by applying these weighted attention scores $S$ to $V$. This method has proven effective for fusing $\mathbf{f}_p$ and $\mathbf{f}_v$. Our supplementary ablation study (Fig.3 Ablation (b)) contrasts this with a direct concatenation of $\mathbf{f}_p$ and $\mathbf{f}_v$, followed by processing through two fully connected layers to produce $\mathbf{f}_g$. The results indicate that direct concatenation yields poor reconstruction performance.
> > >
> > > Although both Points-Net and Vectors-Net could output multiple vector features as a set to be processed through cross-attention, similar to the approaches used in most point cloud segmentation and shape analysis tasks, we opt for a simpler method. Given that we are learning point cloud information from local patches, which contain relatively fewer features, a single-dimensional feature vector is sufficient for our purposes.
> > >
> > > **Q: Siren Network.**
> > > In our response, we discussed integrating our unsupervised method with unsupervised methods to enhance geometric details in reconstructed surfaces. This extension highlights our method's contribution and adaptability. Both CAP-UDF and LevelSetUDF utilize coordinate-based MLPs to predict UDF values, leveraging loss functions such as normal consistency and projection loss to supervise training. Given the proven efficacy of SIREN networks in SDF-based reconstruction tasks, we chose SIREN as our backbone for conducting  integration tests. We initialized the SIREN network using outputs from our method and then proceeded with unsupervised learning. This integration strategy demonstrates that our method can provide a better initialization, leading to improved stability and efficiency in UDF learning. We carried out a comparative experiment on three point clouds shown in Fig.5 in the one-page PDF. We used the same SIREN network and loss functions across tests but differentiated between random initialization and initialization using our method's output. For Tennyson and Horse Head, SIREN with random initialization failed to converge, resulting in poor shape reconstruction. For Gargoyle, while SIREN did converge, we observed that SIREN initialized with our method not only converged faster but also exhibited superior performance in terms of CD and F-scores. We will include detailed implementation on integrating LoSF-UDF with SIREN in revised manuscript.
> > > | Tests | Time(min) | CD($\times 100$) | $F^{0.005}$ | $F^{0.01}$ |
> > > |----------|----------|----------|----------|----------|
> > > | Initialization w/ our method | $\leq$10 | 0.012 | 0.728 | 0.908 |
> > > | Random initialization | 15$\sim$20 | 0.595 | 0.654 | 0.882 |

---

### Official Review · Reviewer_YXLQ · 2024-07-12

**Soundness:** 3
**Presentation:** 3
**Contribution:** 3
**Rating:** 6
**Confidence:** 4

**Summary:**

The paper proposes a novel training strategy to learn Unsigned Distance Fields from local shapes. The idea is to train the model on a dataset of point cloud patches characterized by mathematical functions representing a continuum from smooth surfaces to sharp edges and corners. Although trained only on synthetic surfaces, it demonstrates a remarkable capability in predicting UDFs for a wide range of  surface types. The method is evaluated on “Car” category of ShapeNet dataset in addition to DeepFashion3D, and ScanNet datasets. Furthermore, the paper shows results  on scans from real range scan dataset , in addition to ablation studies highlighting the robustness of the method to noise and outliers.

**Strengths:**

- The paper is well written and easy to follow.
- **Novelty**: While training on local patches is not new, the design of the local shapes and network architecture is novel and effective.
- **Performance**: The paper shows good generalization results while being only trained on local synthetic patches and more robustness to noise and outliers in the input pointcloud.

**Weaknesses:**

- **Inference time**: While the method is better in terms of data storage space, data preparation time and training time, no comparison regarding the  inference time is provided.
- **Patch radius**: The patch radius used at test time is a crucial hyper-parameter for the method. A discussion about how to set this parameter and how it depends on the point cloud density/size would strengthen the paper.

**Questions:**

-  What the inference time of the method compared to other baselines ?
- How to set the patch radius for new shapes with different pointcloud densities and sizes ?

**Limitations:**

The authors adequately addressed the limitations of the method.

---

> ### Author Rebuttal · Authors · 2024-08-07
>
> Thank you for recognizing the contributions of our paper. We also appreciate your valuable suggestions. In the following, we address your main concerns. Please refer to the supplementary one-page PDF for figures and tables.
>
> **Q1: The inference time of the method compared to other baselines.**
> We list the runtime of our method and baseline models in Table.1. All tests were conducted using an Intel i9-13900K CPU and an NVIDIA RTX 4090 GPU-24GB.
>
> **Q2: Setting the radius for new shapes with different point cloud densities and sizes.**
> We provided additional information to confirm the applicability of our method. Please refer to the radius analysis in Fig.1(b) and the point cloud density analysis in Fig.4.
> The choice of radius for extracting local geometries from test data directly influence the complexity of geometric features captured. When point clouds are normalized to a unit bounding box, we set the radius $r=0.018$. This setting achieves satisfactory reconstruction for DeepFashion3D, ShapeNet-Car, commonly used CG models, and several real scans. Also, the bias analysis of local patches between synthetic data and test data in Fig.1 indicates that the results are more reliable when $r=0.018$ because there are fewer outliers. We agree this default setting is not universal. To achieve the best performance for a new dataset, we recommend users conduct a preliminary bias test using a small sample set to fine-tune the radius $r$ appropriately.
> Fig.4 provides reference values for point cloud density within the radius, and our algorithm is not suitable for excessively sparse point clouds. In our future work, We will investigate designing adaptive radii based on local geometric feature sizes and point cloud density.

---

### Official Review · Reviewer_AbEM · 2024-07-12

**Soundness:** 3
**Presentation:** 2
**Contribution:** 2
**Rating:** 4
**Confidence:** 4

**Summary:**

The authors propose a method for open surface reconstruction from 3D point clouds. They train a network to predict unsigned distance functions (UDFs) from point cloud patches using only synthetic data of quadratic surfaces. Evaluation shows that the trained network generalizes well to other complex patterns and is more resilient to noise when reconstructing 3D surfaces from point clouds.

**Strengths:**

The idea of training a UDF regression network using only synthetic data of quadratic surfaces is quite intriguing. This approach allows for a controlled and systematic way to generate training data, which can be more consistent and free from the imperfections and variability found in real-world data.

Using quadratic surfaces as a basis for synthetic data is quite interesting where the authors argue that quadratic surfaces can approximate various local geometries. I have some doubts about this but it is reasonable and novel to a certain extend.

**Weaknesses:**

I have three main concerns: potential biases with the synthetic training data, the evaluation scheme, and the applicability in practice due to the patch radius.


## Potential Biases with Synthetic Training Data:

The use of primitive geometrical patches as training data for a UDF regressor might introduce biases. The observation that any local geometries can be approximated by quadratic surfaces is only valid at a very fine resolution, which requires dense point clouds to observe reliably. It is unclear how many patches have been synthesized and whether they provide a good approximation of universal geometrical primitives. Some analyses would be helpful here: for example, using the ShapeNet car dataset, cropping all local patches from each car, and finding the closest synthesized one to check for approximation errors. Are there any patterns with sufficiently high approximation errors? Can we perform these analyses at different resolutions and see how they correlate with surface reconstruction?

## Evaluation Scheme

The testing data is simulated to match the scenarios the method is designed for: the point cloud is quite dense, and artificial noise is added similarly to the training data. There is no quantitative evaluation for real-world scanned data.

## How sensitive is the method to different patch radii?

The value of 0.018 is oddly specific. I suspect that with a larger value of r, the method will generate overly smooth surfaces (as shown in Figure 6 - right), and with a smaller value of r, it will generate holes due to the point cloud not being dense enough. Overall, there is an inherent issue with this trade-off that may not be resolvable with this approach. Detailed experiments varying the patch radius and analyzing the impact on reconstruction quality would be helpful.

## Additional Concerns

- Ablation studies on the network architecture are missing. I am unsure about the roles of the two branches and the cross-attention mechanism. There is a potential issue with the reliance on Point-Net for embedding computation. Is this network pre-trained on other datasets? If so, we should be careful with the claim of using only synthetic data, as pre-training on real data could influence the results.

- It also seems that the method could be quite slow. The authors should include a speed test to provide insights into the computational efficiency of the proposed approach. Evaluating the method's runtime on different hardware setups and for varying point cloud sizes would give a clearer picture of its practical applicability.

**Questions:**

Main questions I would like the answers for:
- Analyses on using the synthetic patches to approximate the real local 3D geometries.
- Speed test.
- Quantitative results on real-world scanned data, such as 3D-Scene dataset.

**Limitations:**

The authors acknowledged that the proposed method cannot handle incomplete point cloud. However, it is unclear what how resilient it is when dealing with this.

---

> ### Author Rebuttal · Authors · 2024-08-07
>
> Thank you for recognizing the novelty of our method. We also appreciate your valuable suggestions. In the following, we address your main concerns. Please refer to the supplementary one-page PDF for figures and tables.
>
> **Q1: Analyses on using the synthetic patches to approximate the real local 3D geometries.**
> Thanks for your suggestion, we conducted a bias analysis of local patches between our synthetic and test point clouds. First, we extracted the features $\mathbf{f}^i_p$ for all 122,624 patches in our training dataset, processed by Points-Net in our network.
> Then, we extracted all local geometries with a radius $r=0.018$ from point clouds in three categories: DeepFashion3D, ShapeNetCars, and 10 commonly used CG models (e.g., Bynny, Bimba, etc). Subsequently, we obtained the corresponding feature vectors $\mathbf{f}^i_p$ for these local geometries through the same Points-Net. To analyze the bias between our synthetic dataset and the test data, we measured the distances between the feature vectors $\{\mathbf{f}^i_p\}_{i=1}^N$ and $\{\hat{\mathbf{f}}^i_p\}_{i=1}^M$. This analysis was also applied to point clouds with noise (0.25\%) and outliers (10\%). Fig.1 (a) shows the feature bias distribution in box-plots, indicating the spread and skewness of data through the minimum, maximum, median, and Q1, Q3 quartiles. The outliers are samples that significantly deviate from the norm. Overall, the results show a low bias range with an average median of 0.0025.
>
> **Q2: Speed tests for evaluating computational efficiency.**
> We list the runtime of our method and baseline models in Table.1. All tests were conducted using an Intel i9-13900K CPU and an NVIDIA RTX 4090 GPU-24GB. As supervised methods, CAPUDF, LevelsetUDF, and DUDF require extensive UDF learning time. Although their mesh extraction process is relatively fast, the resulting mesh quality is somewhat poor and necessitates normal.
> Comparative results show that supervised, local feature-based methods like GeoUDF and ours outperform unsupervised methods in computational efficiency.
> However, GeoUDF's UDF inference relies on the training dataset's category.
> The evaluation in our manuscript (Sec.4.3) also demonstrates that our method surpasses GeoUDF in terms of generalization, storage complexity, and training time.
>
> **Q3: Quantitative evaluation for real-world scanned data.**
> We illustrate the visual comparison and the quantitative evaluations of different methods on real scan dataset in Fig.2.
>
> **Q4: Sensitivity analysis for patch radii.**
> The choice of radius for extracting local geometries from test data directly influence the complexity of geometric features captured. When point clouds are normalized to a unit bounding box, we set the radius $r=0.018$. This setting achieves satisfactory reconstruction for DeepFashion3D, ShapeNet-Car, commonly used CG models, and several real scans. Also, the bias analysis of local patches between synthetic data and test data in Fig.1 indicates that the results are more reliable when $r=0.018$ because there are fewer outliers. We agree this default setting is not universal. To achieve the best performance for a new dataset, we recommend users conduct a preliminary bias test using a small sample set to fine-tune the radius $r$ appropriately.
>
> **Q5: More ablation studies on the network architecture.**
> To validate the effectiveness of each component, we add two ablation studies: (a) removing the Points-Net and Cross-Attn modules; (b) removing the Cross-Attn module. We tested the performance on noisy point clouds. The results in Figure 3 demonstrate the important role of each component.

---

> > ### Comment · Reviewer_AbEM · 2024-08-12
> >
> > Again, I still find the choice of the radii value oddly specific. I think it is extremely crucial to the method but the current analysis doesn't answer me very well. It does not simply relate the "satisfactory reconstruction" but it hints when the main idea works and when it doesn't. I believe that with different value, the synthetic data will have to be generated differently to better approximate the "universal set". However, if it is too large, no simple function can generate those and if it is too small, the point cloud has to be extremely dense for the method to become relevant.
> >
> > Overall, this is an inherent issue of this approach that makes me find it not very useful.

---

> > > ### Author Response · Authors · 2024-08-13
> > >
> > > Thank you for your feedback. We acknowledge that the quality of LoSF-UDF's reconstruction results depends on the model-dependent hyperparameter $r$, which specifies the size of local patches. However, we do not consider this dependency to be an inherent flaw. If the default setting of $r$ proves suboptimal for a new dataset, our method allows users to fine-tune this parameter using a few representative models. Subsequently, the pre-trained network can be re-run on these models **without the need for re-training**. Because of the high efficiency of LoSF-UDF, this fine-tuning process is both rapid and straightforward.
> > >
> > > It is also important to note that the reliance on a model-dependent hyperparameter is not unique to our method. For instance, GeoUDF (ICCV 2023) employs a hyperparameter $k$ to identify the $k$-nearest neighbors required for computing weights, which represent the contribution of each neighbor. Similarly, POCO (CVPR 2022) also adopts a model-dependent parameter for searching local neighborhoods. To our knowledge, there are no supervised SDF/UDF learning methods that are entirely free of model-dependent hyperparameter.
> > >
> > > Moreover, as demonstrated in the rebuttal, LoSF-UDF can also be effectively combined with an unsupervised method, serving as an initialization. In such scenarios, it is not necessary to obtain a highly accurate result from LoSF-UDF, which diminishes the importance of fine-tuning the parameter $r$.

---

> > > > ### Author Response · Authors · 2024-08-13
> > > > **To continue and supplement**
> > > >
> > > > Following our previous discussion, here we provide a concrete example to demonstrate that the parameter fine-tuning process for LoSF-UDF is both rapid and straightforward. Below, we consider the Bunny model at 5 different resolutions (10K, 15K, 20K, 30K, and 50K) and present LoSF-UDF's reconstruction results across various settings of the hyperparameter $r$:
> > > >
> > > > |··Radius··|····Bunny\_10k····|····Bunny\_15k····|····Bunny\_20k····|····Bunny\_30k····|····Bunny\_50K····|
> > > > |----------|----------|----------|----------|----------|----------|
> > > > |··0.012···|······Failure·····|······Failure·····|·0.029/0.113/0.23·|·0.447/0.729/0.39·|·0.601/0.891/0.52·|
> > > > |··0.018···|·0.101/0.251/0.30·|·0.628/**0.902**/0.53·|·0.816/**0.979**/0.64·|·0.802/**0.998**/0.72·|·0.880/**0.999**/0.81·|
> > > > |··0.024···|·0.537/**0.997**/0.77·|·0.514/0.992/0.87·|·0.599/0.994/0.92·|·0.615/0.992/0.98·|·0.73/0.998/1.09·|
> > > > |··0.036···|·0.407/0.793/1.02·|·0.425/0.700/1.09·|·0.499/0.822/1.19·|·0.600/0.953/1.82·|·0.671/0.982/2.39·|
> > > >
> > > > In this table, we report the F-scores ($F^{0.005}$ and $F^{0.01}$, where higher scores indicate better quality of the reconstruction results) and the running time required for patch extraction and UDF inference (minutes) for each density and radius setting. For each resolution (column), the best F-scores are highlighted using boldface fonts to easily identify the most effective settings of radius $r$.
> > > >
> > > >
> > > > For a very small radius of $r=0.012$, the method was unable to produce valid reconstruction results for the lower resolutions at 10K and 15K densities. The failure is due to the challenge of representing local features when there are extremely few points within the specified radius. The default setting of \( r=0.018 \) yields satisfactory results for inputs with high and medium densities. However, this setting proves inadequate for the lowest density level of 10K points, where it results in fragmented meshes. At this sparse resolution, each patch with a radius of \( r=0.018 \) encompasses merely 5.75 points on average. Adjusting the radius to \( r=0.024 \) increases the average number of points per patch to 9.82, thereby enhancing the reconstruction accuracy for this lower resolution. While increasing the radius further does raise the number of points per patch, performance worsens because the disparity between actual geodesic disks and synthetic patches becomes too pronounced at larger radii. Typically, for a given model with a fixed resolution, the relationship between performance and the parameter $r$ follows an inverted-U-shaped curve. Consequently, **employing a linear search on a few representative models should efficiently determine the optimal hyperparameter $r$ for a new dataset.** We will include both visual results and performance-$r$ plots to illustrate the impact of these parameter adjustments.

---

### Author Rebuttal · Authors · 2024-08-07

Thank you for the valuable feedback. We reiterate our method introduces a novel strategy for learning UDFs from local shape functions. It requires only a single training session on a synthetic dataset composed of local patches represented by simple mathematical functions, and it demonstrates strong generalization capabilities across various point cloud shapes. Our method not only achieves comparable or superior performance to SOTA methods but does so at a significantly lower training cost. Its independence from specific shape categories greatly enhances its applicability. Leveraging its efficiency and generalization capabilities, our method can also integrate seamlessly with unsupervised methods, serving as an effective initialization.

**Q1: Bias Analysis for Local Patches (\#AbEM)**.
We conducted a bias analysis comparing our synthetic local patches with local geometries extracted from test data. We obtained the feature vectors $\mathbf{f}^i_p$ for all patches in the training dataset (totaling 122,624 data points), processed by Points-Net in our network, and then extracted local geometries with a radius $r=0.018$ from point clouds in three categories: DeepFashion3D, ShapeNet-Cars, and 10 common CG models (Bunny, Bimba, etc). We computed the corresponding feature vectors $\hat{\mathbf{f}}^i_p$. We measured the distances between $\mathbf{f}^i_p$ and $\hat{\mathbf{f}}^i_p$ as the bias. This analysis was also applied to point clouds with noise and outliers. Fig.1 (a) shows the feature bias distribution in box-plots. The outliers are samples that significantly deviate from the norm. Overall, the results show a low bias range.
**Joint analysis with radius.** The radius impacts the size of extracted local geometries and thus influences the observed bias. We performed experiments with different radii using the aforementioned analysis method. Fig.1 (b) shows a radius of r=0.018 (our default setting) results in relatively fewer outliers.

**Q2: Detailed illustration of radii and point density (\#AbEM, \#YXLQ, \#1t3W)**. The choice of radius for extracting local geometries from test data directly influences the complexity of the geometric features captured. When normalizing point clouds to a unit bounding box, we set the radius, $r=0.018$. This setting achieves satisfactory reconstruction for DeepFashion3D, ShapeNet-Car and several real scans. Also, the bias analysis shows that this radius setting maintains a relatively low bias, suggesting its effectiveness. We agree this default setting is not universal. To achieve the best performance for a new dataset, we recommend  users perform a preliminary bias test using a small sample set to fine-tune the radius $r$ appropriately.

We also notice the optimal choice of $r$ varies with the sampling rate of the input point cloud. Sparse point clouds, where few points fall within the designated radius, can significantly degrade the quality of the reconstruction results. A possible way for mitigating issues arising from low sampling rates is to apply an upsampling module during the pre-processing step. Fig.4 visually demonstrates the impact of point density on reconstruction results.

**Q3: More explanation and ablation studies on the network architecture (\#AbEM, \#15wp)**.  Our main goal is to derive the UDF value for a query point by learning the local geometry within a radius $r$. To achieve this, we utilize Points-Net to capture the point cloud features $\mathbf{f}_p$ of local patches. This process enables the local geometry extracted from test data to align with the synthetic data through feature matching, even in the presence of noise or outliers. Vectors-Net is tasked with learning the features $\mathbf{f}_v$ of the set of vectors pointing towards the query point, which includes not only the position of the query point but also its distance information. The Cross-Attn module then processes these local patch features $\mathbf{f}_p$ as keys and values to query the vector features $\mathbf{f}_v$, which contain distance information, returning the most relevant feature $\mathbf{f}_G$ that determines the UDF value. See Fig.3 for two ablation studies on noisy point clouds.

**Q4: Quantitative evaluation for real scans (\#AbEM, \#15wp)**. We provide both a visual comparison and  quantitative evaluations of various methods applied to real scans in Fig.2. This illustration demonstrates the effectiveness of our approach compared to competing methods under practical, real-world conditions.

**Q5: Time efficiency (\#AbEm, \#YXLQ, \#15wp)**.
Table 1 presents the runtime for our method, including local patch extraction, network inference, and mesh extraction using DCUDF. All tests were conducted on an Intel i9-13900K CPU and an NVIDIA RTX 4090 GPU. The runtime for local patch extraction and UDF inferring varies depending on the radius setting, which influences the number of effective query points.
Computational results show that supervised, local feature-based methods like GeoUDF and our approach significantly outperform unsupervised methods in terms of computational efficiency.
However, GeoUDF's dependence on the training dataset's category limits its applicability. Furthermore, the evaluation detailed in Sec 4.3 reveal our method exceeds GeoUDF in generalization capabilities, storage requirement, and training time efficiency.

**Q6: Integration of LoSF-UDF with unsupervised methods (\#15wp)**
Due to the high efficiency and generalizability of our method, it serves as an effective initialization for unsupervised methods. We train a Siren network using UDF values derived from our pre-trained LoSF-UDF for supervision, employing MSE, Eikonal and normal alignment losses. The initialization provided by our method not only stabilizes the network but also accelerates its convergence, cutting the training time from 30 minutes to 10 minutes. Fig.5 showcases 3 examples where this strategy significant enhances geometric accuracy and fidelity.

---

### Comment · Area_Chair_m1tP · 2024-08-07

Hi reviewers,

Thanks a bunch for all your hard work as reviewers for NeurIPS 2024.

The discussion period between reviewers and authors has started. Make sure to check out the authors' responses and ask any questions you have to help clarify things by 8.13.

Best,
AC

---

> ### Comment · Area_Chair_m1tP · 2024-08-13
>
> Dear reviewers,
>
> As the reviewer-author discussion period is about to end by 8.13, please take a look at other reviewers' reviews and authors' rebuttal at your earliest convenience. It would be great if you could ask authors for more clarification or explanation if some of your concerns are not addressed by the rebuttal.
>
> Thanks,
>
> AC

---

### Author Response · Authors · 2024-08-14
**Thank You for Your Feedback**

Dear Reviewers,

Thank you for taking the time to review our responses and for providing additional feedback. We would like to take this opportunity to highlight the unique contributions of our paper.

Unlike existing supervised learning methods for UDFs, which are often tailored to specific training datasets and may require re-training for new datasets, LoSF-UDF does not exhibit such dataset dependence. Once trained on our synthetic datasets, it can be applied directly to new datasets without modification. Our method involves only one model-dependent parameter—the size of local patches. With the default setting, we tested our approach on five diverse datasets: DeepFashion3D, ShapeNetCars, scanned data from SRB and threedscans, and scene point clouds, all of which yielded good experimental results.

If the default setting leads to suboptimal results on new datasets, users can easily fine-tune this hyperparameter through a simple linear scan, without the need for re-training. Given LoSF-UDF's high runtime performance, enhanced robustness against noise and outliers, dataset independence, and potential for integration with unsupervised methods, we believe it can be applied to a wide range of real-world applications.

Thank you once again for your valuable feedback.

Sincerely,
The Authors

---

### Decision · Program_Chairs · 2024-09-25

**Decision:**

Reject

**Comment:**

The submissions got two positive recommendations and two negative recommendations. The reviewers were concerned about the sensitivity to parameters, computational complexity, and method justification. The authors addressed some of these concerns, but some reviewers were still concerned about the limited scenarios where the proposed method can work. During the AC and reviewers discussion period, the AC and the reviewers also had intense discussions on the novelty over some previous methods which also used patches from meshes rather than the ones produced by equations. According to these discussions, it seems that the proposed method is not novel enough. Regarding this,  the AC did not see any explanations on the difference to the related works  in the authors' reply to reviewer 1t3W’ s questions either. Per these, the AC made a decision to reject this submission. The decision was approved by the SAC.